# The Design, Synthesis and Application of Nitrogen Heteropolycyclic Compounds with UV Resistance Properties

**DOI:** 10.3390/ijms24097882

**Published:** 2023-04-26

**Authors:** Biao Yang, Xinbo Yang, Yuchuan Li, Siping Pang

**Affiliations:** 1School of Materials Science and Engineering, Beijing Institute of Technology, Beijing 100081, China; 2School of Mechatronical Engineering, Beijing Institute of Technology, Beijing 100081, China

**Keywords:** design, synthesis, characterization, ultraviolet absorption, ageing resistance

## Abstract

Exposure to ultraviolet (UV) light is known to cause skin aging, skin damage, cancer, and eye diseases, as well as polymer material aging. Therefore, significant attention has been devoted to the research and development of UV absorbers. Considering the robust hydrogen bonding and conjugated structure present in nitrogen-containing polycyclic compounds, these compounds have been selected as potential candidates for exploring ultraviolet absorption properties. After structural optimization and the simulation of ultraviolet absorption spectra, four tris-[1,2,4]-triazolo-[1,3,5]-triazine (TTTs) derivatives, namely TTTB, TTTD, TTTJ, and TTTL, were selected as the preferred compounds and synthesized. The structure of the compound was determined using various analytical techniques, including FTIR, ^1^HNMR, ^13^CNMR, HRMS, and XRD. Subsequently, composite films of polyvinyl chloride (PVC) and TTTs were produced using a simple solvent casting technique. The PVC films were subjected to UV age testing by exposing them to an ultraviolet aging chamber. The age-resistant performance of the fabricated films was evaluated using an ultraviolet spectrophotometer and Fourier infrared spectrum instrument. The findings suggest that TTTs exhibit a noteworthy capacity for absorbing ultraviolet radiation. The TTTL compound exhibits a superior UV absorption performance compared to commercially available UV absorbers such as UV-0 and UV-327 in the market.

## 1. Introduction

With the intensification of human activities leading to the depletion of the ozone layer, more ultraviolet (UV) radiation is penetrating the atmosphere and reaching the Earth’s surface, posing serious threats to both living organisms and the environment [1,2]. Overexposure to UV radiation has been linked to various health issues, including skin aging, damage, cancer, and eye diseases, collectively posing a significant threat to public health [3,4]. Additionally, UV radiation can negatively impact plant and animal life, causing the thinning of plant leaves, and reduced growth and yields, ultimately affecting crop production [5]. It can also impact animal vision, reproductive capacity, appetite, and behavior patterns [6]. However, the negative effects of UV radiation are not only limited to living organisms, as they can also impair the physical and chemical properties of polymer materials. Exposure to UV radiation can cause the polymer chains to break down, resulting in the formation of free radicals that can react with other polymer molecules, leading to chain scission and the formation of smaller molecular fragments. This process can weaken the polymer’s physical and chemical properties, such as its tensile strength, ductility, and thermal stability, and can result in discoloration, cracking, and the overall deterioration of the material [7,8]. Ultraviolet radiation accelerates the degradation of polymers, leading to a decrease in their efficiency and limiting their potential applications in various fields. The aging failure of polymer materials has become a crucial issue that restricts their further development and application [9,10]. Therefore, addressing this issue is imperative to ensure the sustainable and safe use of polymer materials in various domains.

To address this issue, a variety of UV absorbers have been exploited to mitigate the hazards of UV radiation. In terms of the UV absorption mechanism, most UV absorbers can be divided into two types. The first type involves the fracture and recombination of hydrogen bonds, such as benzophenones, benzotriazoles, and triazines [11]. The UV absorption properties of these compounds can be attributed to the fracture and recombination of intermolecular and intramolecular hydrogen bonds at UV wavelengths, leading to the release or consumption of ultraviolet energy as low-energy radiation [12,13]. Another mechanism involves the Photo-Fries rearrangement, which produces a compound with UV absorption ability, such as salicylate. These compounds do not possess an intrinsic UV absorption capacity, but their structures, generated through the Photo-Fries rearrangement, can reduce or eliminate UV radiation damage by releasing or consuming UV energy as low-energy radiation [14]. Researchers have made significant progress in developing new and more effective UV absorbers. For example, Chen et al. developed triazine UV absorbers with surface enrichment characteristics to improve their distribution on coating surfaces [12]. Xu et al. synthesized bisbenzotriazole (DSB-HTBB) to protect polymer materials and sensitive skin from UV rays [13]. In another study, M.A. Sangamesha et al. investigated the UV absorption properties of 4-Hydroxy Benzophenone (Ph_2_CO) and applied it to cosmetics and plastics [7]. They also developed novel fluorinated triazine UV absorbers with surface enrichment properties. However, the UV absorbers available on the market currently have relatively limited functionality and may potentially bioaccumulate, thereby disturbing the hormonal balance in organisms. If these chemicals continue to be released into the natural environment and accumulate to a certain level, they could potentially disrupt the ecological balance [15,16]. Moreover, UV absorbers can have detrimental effects on human health, as they may adhere to the skin and cause allergies, dermatitis, and other adverse reactions. Therefore, the development of more efficient and antibacterial UV absorbers while reducing their harmful impact on the ecosystem is needed [17].

In recent years, there has been significant interest in fused ring triazole compounds due to their strong intramolecular and intermolecular hydrogen bonds, excellent stability, good compatibility, and certain bactericidal properties [18,19,20]. These compounds introduce additional heterocyclic ring formations on the triazole ring structure, providing more high-energy chemical bonds such as C-N, N-N, N=N, and N-O, as well as a larger conjugated structure and more chemical modification sites compared to monocyclic and cyclitic energetic compounds [21,22,23]. As a result, fused ring triazole compounds have found wide applications in fields such as energy, medicine, and liquid crystals [24,25,26]. The molecular structure of triazoles features high conjugation and intramolecular hydrogen bonding, which endows them with strong UV-absorbing capabilities. Additionally, triazole molecules possess high stability and chemical activity, making them suitable as UV absorbers in various polymeric materials [27]. Consequently, the triazole structure has become one of the focal points for researching and developing new types of UV absorbers.

In this study, we designed and synthesized four derivatives of tris-[1,2,4]-triazolo-[1,3,5]-triazine (TTTs), namely tris-phenyl-[1,2,4]-triazolo-[1,3,5]-triazine (TTTB), tris-(2-hydroxyphen-1-phenyl)-[1,2,4]-triazolo-[1,3,5]-triazine (TTTL), tris-(4-hydroxyphen-1-phenyl)-[1,2,4]-triazolo-[1,3,5]-triazine (TTTJ), and tris-(4-hydroxyphen-1-phenyl)-[1,2,4]-triazolo-[1,3,5]-triazine (TTTD). This tris-[1,2,4]-triazolo-[1,3,5]-triazine derivatives, which comprise a polycyclic conjugated structure with a 1,2,4-triazole ring and a 1,3,5-triazine ring [25]. The π–π interactions between its molecules exhibit high stability, while the hydrogen atoms on the triazole ring can form both intramolecular and intermolecular hydrogen bonds. The breaking and reformation of hydrogen bonds can convert the energy of ultraviolet radiation into heat or other forms of non-radiative energy, effectively protecting the substance from UV damage. We optimized the molecular structure of TTT compounds and analyzed the effects of both intermolecular and intramolecular hydrogen bonding. Additionally, we calculated and simulated the UV absorption spectra of TTT compounds to elucidate their photophysical properties. Considering the broad application of polyvinyl chloride (PVC) as a representative of general-purpose plastics and its poor aging resistance, we selected PVC as the test material in our experiments to evaluate the efficacy of TTTs, as UV absorbers to improve their aging resistance [28,29,30]. Specifically, we added UV absorber TTTs to PVC and produced a film using a solvent-pouring method. We then subjected the PVC film to ultraviolet aging in the UV aging tester and subsequently evaluated the film’s performance.

## 2. Results and Discussion

### 2.1. Electronic Structure

The interaction region indicator function (IRI) [31,32] can clearly demonstrate various interactions within molecules. Figure 1 illustrates the IRI isosurface, highlighting interactions within the TTTs series of molecules. Unlike TTTJ and TTTD, the TTTL molecule contains a neighboring hydroxyl group, which can form an intramolecular hydrogen bond between adjacent triazole rings with a stronger hydrogen bond interaction. When the molecule is irradiated with UV light, the electrons in the molecule jump to higher energy levels, while the intramolecular hydrogen bonds break, leaving the entire structure in an unstable high-energy state. The absorbed UV energy is then converted into other forms of energy and released, followed by the electron returning to the ground state and the intramolecular hydrogen bond’s reclosing. Therefore, TTTL has the strongest UV absorption performance. This observation suggests that TTTL compounds with neighboring hydroxy groups possess superior UV absorption properties. This provides indirect evidence for the interplay between the proximal -OH moiety and the benzene ring, which contributes to enhanced ultraviolet absorption characteristics [33].

To directly investigate the influence of TTTs’ hydrogen bond interaction on ultraviolet absorption spectra, Fingerprint plots, and Hirshfeld surfaces were employed to analyze the hydrogen bond in crystals (Figure 2). “*d*_e_” refers to the distance from a point on the surface to the nearest nucleus outside the surface [32]. Figure 2 depicts the hydrogen bond interaction proportionate to the overall interaction within the crystal. It was observed that when the benzene ring lacked a -OH bond, the proportion of hydrogen bond interactions in the crystal was 23.6%, whereas when the benzene ring contained the -OH bond, the proportion of hydrogen bond interactions in the crystal was 33.4%. The overall hydrogen bond effect was improved by approximately 10%, which directly demonstrates how the presence of more hydrogen bond interactions in the system was the reason behind the good ultraviolet absorption performance of triazole and triazine-containing ortho-hydroxybenzene ring structures, which is consistent with the research findings of Zhang et al [34]. 

In order to better understand the structures and properties of TTTB, TTTD, TTTJ, and TTTL molecules, the HOMO and LUMO energies of these compounds were predicted, and schematic diagrams of their respective LUMO and HOMO orbitals were drawn. Figure 3 shows the distribution and energy positions of the HOMO and LUMO orbitals of TTTB, TTTD, TTTJ, and TTTL within the molecules. The HOMO energy levels of TTTB, TTTD, TTTJ, and TTTL were −7.055 eV, −6.431 eV, −6.651 eV, and −6.397 eV, respectively. The corresponding LUMO energy levels were −1.738 eV, −1.499 eV, −1.666 eV, and −2.346 eV. The HOMO orbital of TTTB was distributed throughout the entire molecule, while the LUMO orbital was not distributed on one of the benzene rings. The HOMO orbitals of TTTD and TTTL were mostly distributed on the benzene rings, while the LUMO orbitals were mostly distributed on the tris-[1,2,4]-triazolo-[1,3,5]-triazine ring system. The HOMO orbital of TTTJ was mainly concentrated on one of the benzene rings, while the distribution of the LUMO orbital was shifted towards the tris-[1,2,4]-triazolo-[1,3,5]-triazine ring system. The distribution of frontier orbitals and energy levels for four different molecules suggests that these molecules may exhibit different chemical reactivity and charge transfer phenomena. The frontier orbital gaps of TTTB, TTTL, TTTJ, and TTTD were 5.317 eV, 4.050 eV, 4.985 eV, and 4.932 eV, respectively. The lowest frontier orbital gap of TTTL indicates that it has higher chemical reactivity and is more likely to undergo electronic transitions.

The TD-DFT method was employed to simulate the ultraviolet absorption spectra of the Poisson–Boltzmann solvation model (PBF) and solvation models in various solvents at the calculated PBE0-D3/6-311+G** level, as shown in Figure 4. The results indicate that TTTB, TTTD, TTTJ, and TTTL exhibited broad UV absorption in the range of 200–320 nm. Moreover, to explore the impact of different solvents on the UV absorption of TTTs, the UV absorption spectra of TTTL were simulated in various solvents such as methanol, butanone, acetone, ethanol, and dichloromethane, among others. As illustrated in Appendix A, the five UV absorption lines almost overlapped, indicating that different solvents had a negligible impact on the UV absorption of the UV absorber TTTL. 

### 2.2. Characterization of TTTs

#### 2.2.1. Synthesis and Characterization

Although some synthetic methods have been reported for the tris-[1,2,4]-triazolo-[1,3,5]-triazine class of compounds, most of them were difficult to synthesize and had low yields. In 2016, Alexandre et al. proposed a method for synthesizing tris-(4-hydroxyphen-1-phenyl)-[1,2,4]-triazolo-[1,3,5]-triazine that required the reaction to be carried out under argon protection [35]. However, this synthetic method has stringent conditions, and the yield is only 60%. We attempted to use this method to synthesize TTTL and TTTJ, but we were unable to obtain the target compounds. By optimizing the synthesis scheme and changing the reaction conditions, including increasing the reaction temperature, changing the starting materials and solvents, and modifying the post-treatment methods, we successfully synthesized the target compounds TTTL, TTTJ, and TTTD with yields of 78%, 75%, and 80%, respectively, while also reducing the reaction time. For example, to obtain the target products, we continuously increased the reaction temperature during the cyclization process and tried different solvents, such as methanol, butanone, DMF, ethanol, and DMSO. We also attempted various post-treatment methods, including extraction, recrystallization, rotary evaporation, and column chromatography. The structures of TTTB, TTTD, TTTJ, and TTTL were confirmed through various testing methods such as FTIR, ^1^H NMR, ^13^C NMR, and HRMS. Furthermore, we obtained single crystals of TTTB and TTTL and analyzed their structures using XRD. Overall, the optimized synthesis scheme and conditions allowed us to improve the yield and shorten the reaction time, leading to the successful synthesis of target compounds.

#### 2.2.2. Nuclear Magnetic Resonance Analysis

The compounds TTTB, TTTD, TTTJ, and TTTL were synthesized and dissolved in deuterated dimethyl sulfoxide for ^1^H and ^13^C NMR analysis using a 400 MHz NMR spectrometer. The spectra are shown in Appendix A (see Appendix A). In TTTB, the chemical environment of the hydrogen atoms was very similar, with absorption peaks at 7.75–7.57 ppm and 8.00 ppm corresponding to the H atoms on the benzene ring, and absorption peaks at 125.12 ppm, 129.01 ppm, 130.31 ppm, and 131.94 ppm corresponding to the C atoms on the benzene ring. The absorption peak at 142.40 ppm corresponded to the C atom on the triazole, and the peak at 149.59 ppm corresponded to the C atom on the triazine ring. These results confirm the successful synthesis of TTTs fused in the tricyclic structure. In TTTL, the absorption peaks at 10.31 ppm and 160.00 ppm corresponded to the H atoms in the hydroxyl group, confirming the presence of the hydroxyl group. The other compounds exhibited similar peak positions to TTTL. Taken together, these results clearly demonstrate the successful synthesis of the desired TTTs compounds.

#### 2.2.3. Infrared Spectrum Analysis

The present study reports the successful synthesis of compounds TTTB, TTTL, TTTJ, and TTTD, as demonstrated by their characteristic vibrational spectra. The TTT compounds were subjected to infrared spectroscopy using the potassium bromide pressing method for analysis. The FTIR spectra of TTTB, TTTD, TTTJ, and TTTL (Figure 5a) were analyzed to identify primary functional groups in the UV absorbers. In particular, the C-N and C=N stretching vibration absorption peaks observed on the triazine and triazole rings provided compelling evidence for the successful construction of a fused tris-[1,2,4]-triazolo-[1,3,5]-triazine skeleton. TTTB was characterized by a strong C-N stretching vibration absorption peak at 1602 cm^−1^ on the 1,3,5-triazine ring, along with C-N and C=N stretching vibration absorption peaks at 1481 cm^−1^, 1313 cm^−1^, and 1191 cm^−1^ on the 1,2,4-triazole ring. On the other hand, the remaining three compounds, TTTL, TTTJ, and TTTD, exhibited a distinct OH absorption peak at 3413 cm^−1^, 3411 cm^−1^, and 3381 cm^−1^, respectively, on the benzene ring. These compounds also showed a C-N stretching vibration absorption peak at 1614 cm^−1^, 1612 cm^−1^, and 1614 cm^−1^ on the 1,3,5-triazine ring. Compound TTTL displayed stretching vibration absorption peaks of the C-N and C=N bonds on the 1,2,4-triazole ring at 1458 cm^−1^, 1384 cm^−1^, and 1226 cm^−1^, respectively. Compound TTTJ and TTTD exhibited absorption peak positions on the 1,2,4-triazole ring that were similar to those of TTTL. These observations provide insights into the molecular structures and functional groups present in these compounds. The higher C-N stretching vibration absorption peak observed in TTTB was consistent with the presence of the triazine ring in this compound. The OH absorption peak observed in TTTL, TTTJ, and TTTD was indicative of hydroxyl groups present on the benzene ring. Moreover, the position and intensity of the C-N and C=N stretching vibration absorption peaks in each compound reflect differences in their molecular structures and functional groups.

#### 2.2.4. Ultraviolet-Visible Spectrum Analysis of TTTs

The UV absorption spectra of TTTB, TTTD, TTTJ, and TTTL are shown in Figure 5b. The number of moles in the UV absorption center of each compound solution was kept constant during the test. From the graph, it can be observed that the four types of ultraviolet absorbers, TTTB, TTTD, TTTJ, and TTTL, had wide absorption bands between 200 nm and 400 nm. TTTB exhibited a strong absorption between 230 nm and 290 nm, which was due to an increase in electron density in the conjugated system. Its absorption ability was weakened in the region beyond 300nm. By contrast, TTTD, TTTJ, and TTTL showed two broad absorption peaks, exhibiting a strong absorption between 250 nm and 360 nm. Unlike TTTB, TTTD, TTTJ, and TTTL molecules contain hydroxyl groups in their molecular structure, which exhibit strong electron-attracting and electron-donating properties. This alters the distribution of electron density within the molecule, resulting in changes to its energy state and causing the broadening of absorption peaks and increased absorption intensity. In addition, the scaffold structure of tris-[1,2,4]-triazolo-[1,3,5]-triazine connects the triazole ring and triazine ring in a conjugated system, forming a larger conjugated structure. The conjugated structure can increase the extension of the electron cloud of the molecule, improve the resonance stability of the electrons, and affect the energy level distribution of π electrons in the molecule, resulting in an enhancement of the absorption peak intensity. This is also the reason why the experimentally measured UV absorption spectra differed from the predicted spectra obtained by theoretical calculations, making the absorption peaks in the actual UV absorption spectrum broader and stronger. Therefore, it can be inferred that all four ultraviolet absorbers are capable of effectively absorbing ultraviolet rays and possess an excellent ultraviolet absorption capacity.

#### 2.2.5. Single Crystal X-ray Diffraction Pattern

Pure TTTB and TTTL were obtained through silica gel column purification from crude products. The products were dissolved in a mixed solvent of dichloromethane and methanol (3:1) and were slowly evaporated at room temperature, resulting in pale yellow and colorless transparent crystals of TTTB (CCDC2257622) and TTTL (CCDC2257623), respectively. The TTTB crystal was resolved at 100 K with a 50% probability of ellipsoid thermal displacement. Similarly, the TTTL crystal was resolved at 193 K with a 50% probability of ellipsoid thermal displacement (Appendix A). In the fused ring structures of compounds TTTB and TTTL, the bond lengths in the cyclic structure ranged from 1.29 to 1.40 Å, and the dihedral angle between the triazole and the triazine rings did not exceed 10°. The structures of TTTB and TTTL were further determined. As shown in Appendix A and Figure 6, the TTTB crystal belongs to the *P-1* space group of a monoclinic system, with two molecules per unit cell (*Z* = 2) and a unit cell size of *V* = 954.69(5) Å. The crystal density was determined to be 1.494 g cm^−3^ at 298 K. The TTTL crystal belongs to the *C2/c* space group of a monoclinic system, with eight molecules per unit cell (*Z* = 8) and a unit cell size of *V* = 4901.3(9) Å. The crystal density was determined to be 1.489 g cm^−3^ at 298 K. The molecules form a wavy stacking structure had an interlayer distance of 3.596 Å between adjacent parallel molecules. In the crystal structure of TTTB, intramolecular hydrogen bonds could be formed between the benzene ring and the triazole ring, but the interaction was relatively weak. Similarly, the intermolecular interaction between two TTTB molecules is also weak. In contrast, in the crystal structure of TTTL, strong intramolecular hydrogen bonds could be formed between the adjacent -OH and triazole ring, which could facilitate the release of ultraviolet energy through their breaking and recombination, which is consistent with our theoretical calculations. However, as the TTTL crystal we obtained contained water molecules, the -OH group preferentially formed hydrogen bonds with water molecules instead. 

### 2.3. Characterization of Thin Film

#### 2.3.1. Infrared Spectrum Analysis of Thin Films

FTIR was used to characterize four types of PVC/TTTs films. As shown in Figure 7a, the absorption peak at 2915 cm^−1^ can be attributed to the stretching vibration of -CH-, and the peak at 1424 cm^−1^ can be attributed to the absorption peak of CH_2_. The peak at 956 cm^−1^ corresponds to the stretching vibration of C-C. The absorption peak at 607 cm^−1^ is the absorption peak of the C-Cl group. These absorption peaks are the infrared absorption peaks of the main functional groups of PVC. The characteristic absorption peak of -OH was detected at 3352 cm^−1^, and the stretching vibration absorption peaks of C-N and C=N on the triazole ring were observed at 1384 cm^−1^, 1333 cm^−1^, and 1090 cm^−1^, respectively. The antisymmetric stretching vibration absorption peak of C-N on the triazine ring was found to be 1672 cm^−1^. The absorption peaks of various functional groups in the film were identified, including the main functional groups of PVC and the functional groups of TTTs. These findings indicate that even at low concentrations, UV absorbers can be detected in the film, and TTTs were successfully added to the PVC film.

#### 2.3.2. Carbonyl Index (CI) of Films

The carbonyl index (CI) was used to evaluate the degradation degree of the PVC film [36]. CI was calculated as follows: $$\mathrm{CI}=\frac{Ac}{Ar}$$

*Ac* is the absorbance of the carbonyl vibration peak at 1715 cm^−1^, and *Ar* is the absorbance of multiple -CH_2_- groups at 729 cm^−1^. 

After being exposed to UV radiation, PVC underwent oxidation reactions that resulted in the formation of carbonyl groups on the polymer chain. The formation of carbonyl groups resulted in the appearance of a new peak around 1715 cm^−1^ in the FTIR spectrum, which was not present before UV irradiation, and the intensity of this peak increased with the increasing irradiation time. The carbonyl index (CI) is commonly used as the main indicator of photodegradation and can be defined as the ratio of the absorbance of the C=O peak (around 1715 cm^−1^) to that of the -CH_2_- peak (around 729 cm^−1^). The CI is a measure of the relative absorbance of the carbonyl group compared to a reference band and decreases when low molecular weight components are formed and transferred. Figure 7b shows the decrease in CI for PVC and PVC/TTTs films. The broken lines in the figure show that the CI values for PVC/TTTB, PVC/TTTD, PVC/TTTJ, and PVC/TTTL films all decreased, with the PVC/TTTB film exhibiting the most significant decrease. Following 10 h of irradiation, the CI values for PVC, PVC/TTTB, PVC/TTTD, PVC/TTTJ, and PVC/TTTL films decreased by 22.0%, 13.8%, 14.1%, 15.3%, and 13.0%, respectively. After 20 h of ultraviolet irradiation, the CI values of PVC and PVC/TTT films decreased slowly. Furthermore, compared to pure PVC films, the CI values of PVC/TTT films decreased by less than 5% after 40 h of ultraviolet irradiation. Throughout the aging period, CI decreased in the following order: pure PVC > PVC/TTTJ > PVC/TTTD > PVC/TTTB > PVC/TTTL. These results indicate that TTTs exhibit high resistance to aging.

#### 2.3.3. Effect of Mass Fraction of UV Absorber 

Given its common use in various applications and its high susceptibility to degradation, PVC has been selected as the primary focus of our research in developing UV-absorbing agents [5,37]. To investigate the impact of UV absorber mass fraction on the anti-aging ability of PVC and PVC/TTT films, samples with 0.5 wt%, 1.0 wt%, and 1.5 wt% were prepared. The UV-visible transmittance spectra of pure PVC and the prepared PVC/TTT films were measured in the wavelength range of 200 nm–600 nm, and the results are presented in Figure 8. Notably, pure PVC exhibits high transmittance in the 200 nm–600 nm region, while PVC/TTT films have a negligible UV transmittance below 400 nm, and their transparency exceeds 85% above 400 nm, indicating excellent transparency in the visible light range. The incorporation of TTTs into PVC results in a substantial decrease in the UV transmittance of PVC/TTT films, and the UV absorption range of the films expanded with increasing TTT mass fraction. As the mass fraction of TTTs increased, the transmittance of PVC/TTT films above 400 nm gradually declined, and that of 1.5 wt% PVC/TTTs fell below 80%, thereby impacting the transparency of the PVC film. These findings suggest that PVC films with an appropriate TTT mass fraction can effectively absorb UV rays and provide anti-aging properties while maintaining adequate transparency in the visible light range. While 1.5 wt% PVC/TTT films exhibit stronger UV absorption than those with 1.0 wt%, it is crucial to consider good film transparency as a key factor. Therefore, a 1.0 wt% content of TTTs was deemed the most appropriate. Additionally, for comparison, two of the best ultraviolet absorbers currently available on the market, namely UV-0 (2,4-Dihydroxybenzophenone) and UV-327 (2-(2′-hydroxy-3′,5′-di-tert-butylazole)-5-chlorobenzotriazole), were also tested against TTTs. The results indicate that the mass fraction of UV-0 and UV-327 had a similar effect on PVC films as TTTs. Hence, a content of 1% of the UV absorber was utilized in subsequent tests.

#### 2.3.4. Ultraviolet-Visible Absorption Spectra of Thin Films

The UV-vis spectra of four PVC/TTT films are presented in Figure 9, where the absorption range and maximum absorption wavelength are critical parameters for assessing the UV absorption capability. Specifically, the absorption of films at 200–400 nm was examined in this study. The absorbance values of TTTL, TTTJ, TTTD, and TTTB were 1.88, 1.78, 1.44, and 1.58, respectively, while the absorbance of pure PVC was 0.96. The results show that the absorbance of PVC/TTTL, PVC/TTTJ, PVC/TTTD, and PVC/TTTB films increased by 95.8%, 85.4%, 50.0%, and 64.6%, respectively, compared with the pure PVC film. To investigate the anti-aging properties of PVC and PVC/TTT films, samples were placed in an ultraviolet aging chamber to accelerate ultraviolet irradiation, and their properties were studied after irradiation for 2 h, 6 h, 10 h, 20 h, 30 h, and 50 h. As the irradiation time increased, the UV absorption properties of the PVC and PVC/TTT films gradually decreased, but the rate of decline slowed down after 15 h. This indicates that PVC films with ultraviolet absorber TTTs can absorb ultraviolet rays more effectively and have an anti-aging effect.

To investigate the practical application effect of synthesized UV absorber TTTs, the UV absorber TTTL was compared with the commercially available UV absorbers UV-0 and UV-327. As illustrated in Figure 9, the absorbance of PVC/TTTL, PVC/UV-0, and PVC/UV-327 films were found to be 1.88, 1.42, and 1.74, respectively. Compared to the pure PVC film, the UV absorbance of PVC/TTTL, PVC/UV-0, and PVC/UV-327 films increased by 95.8%, 47.9%, and 81.2%, respectively. The graph demonstrates how PVC/TTT films exhibited a superior UV-blocking ability and transparency, thus possessing high practical value. When compared to commonly used UV absorbers such as UV-0 and UV-327, TTTs demonstrate remarkable performance, featuring a wider range of UV absorption and a stronger capacity for absorption. In contrast, TTTJ’s performance is on par with that of UV-0 and UV-327. These findings indicate that the addition of UV absorber TTTs can effectively reduce the aging rate of PVC films. Overall, the results suggest that the use of TTTs is a promising approach for enhancing the performance of PVC films in UV-blocking applications.

#### 2.3.5. Visual Analysis of the Films in the UV Aging Box

The degradation of PVC films and four PVC/TTT films at different aging times are depicted in the pictures presented in Table 1 and Appendix A. Based on the power of the UV aging chamber and the distance between the samples and the light source, 1 h of accelerated UV irradiation in the aging chamber is equivalent to 100 h of outdoor exposure under normal conditions [38,39]. The pure PVC film showed a burnt yellow color after 48 h, followed by turning into dark brown after 96 h and exhibiting a large area of fracture, while the color became dark after 144 h. The aging degree of the PVC/TTTB film was better than that of the pure PVC film, but this effect was not significant. Compared with UV-0 and UV-327, the anti-aging effect of PVC/TTTD and PVC/TTTJ films was comparable to that of the market. The UV absorption effect of PVC/TTTL films was excellent, and their anti-aging ability was strong. After 144 h of irradiation, only a part of the yellow phenomenon appeared, and the film’s integrity remained good. The aging rate of the pure PVC film was the fastest, and the addition of the UV-absorbing TTTL to the PVC film significantly reduced the aging rate, maintaining the film’s integrity and reducing color change. These findings suggest that synthetic TTTs can serve as an additive with an excellent UV-blocking ability.

## 3. Materials and Methods

### 3.1. Calculation Method

The Gaussian 09 program was employed to systematically calculate DFT using the PBE0-D3 density functional. Geometric structure optimization and frequency calculations were performed using the theoretical level of PBE0-D3/6-311G**. Furthermore, the TD-DFT method was utilized to simulate the ultraviolet absorption spectra of PBF solvation models in different solvents, including the gas phase, at the calculation level of PBE0-D3/6-311+G** [40,41,42]. To visualize the interactions present in the designed molecules, Multiwfn 3.8 [43] was used to perform molecular frontier orbital and interaction region indicator (IRI) [32] analyses on TTTs, while CrystalExplorer 17.5 [44] was used to conduct Hirshfeld [31] surfaces and Fingerprint plot analyses. 

### 3.2. Materials

All reagents, including benzonitrile, 2-cyanophenol, NH_4_Cl, 3-cyanophenol, 4-cyanophenol, DMF, anhydrous K_2_CO_3_, ethyl acetate, petroleum ether, and sodium hydroxide, were purchased from Aladdin (Shanghai, China) and were of analytical grade. Sodium azide, of an analytical grade, was purchased from Tianjin Fengchuan Chemical Reagent Technology Co., Ltd. (Tianjin, China). A total of 36–38% hydrochloric acid, acetic anhydride, methanol, butanone, dichloromethane, and acetone were analytical graded and provided by Beijing Tongguang Fine Chemicals Co., Ltd. (Beijing, China). Potassium bromide was of chromatographic grade and purchased from China National Pharmaceutical Group Chemical Reagent Co., Ltd. (Beijing, China). All reagents were used as received without further purification. Infrared (IR) spectra were acquired using a Thermo Nicolet AVA-TAR 370 FT-IR spectrometer. ^1^H NMR and ^13^C NMR analyses were performed on an Advance Bruker III 400 (400 MHz) NMR spectrometer, with TMS as the internal standard and DMSO-d_6_ as the solvent. UV-vis spectroscopy was carried out using a Hitachi U4150 ultraviolet spectrometer in the 200–600 nm range, with methanol were used as the solvent for sample analysis. The separation of compounds was accomplished by High-Resolution Mass Spectrometry (HRMS) utilizing a Triple Quadrupole MS instrument (Waters India Pvt Ltd., Karnataka, India).

### 3.3. Preparation of TTTB, TTTD, TTTJ and TTTL

The synthetic routes of TTTB, TTTD, TTTJ and TTTL are shown in the Figure 10, respectively.

Synthesis of TTTB. In a 50 mL three-neck flask, a sequence of 0.618 g (6 mmol) of benzonitrile, 0.963 g (18 mmol) of NH_4_Cl, 25 mL of DMF, and 1.17 g (18 mmol) of sodium azide was added. The mixture was heated to 80 °C for 1 h and then to 125 °C for 10 h. After the reaction was complete, the solution was quenched in 100 mL of water and acidified with 6 mol/L HCl to pH = 1. The resulting product, 5-phenyltetrazole, was obtained by recrystallization in water and filtration. Next, a 100 mL three-neck flask was charged with 1.46 g (9 mmol) of 5-phenyltetrazole and 60 mL of methanol, and the mixture was stirred at room temperature until 5-phenyltetrazole was completely dissolved. Next, anhydrous K_2_CO_3_ (4.14 g, 30 mmol) and cyanuric chloride (0.55 g, 3 mmol) were added to the reaction mixture and allowed to react for 20 h at 90 °C. The filtrate was collected for spin steaming, and the resulting product, tris-phenyl-[1,2,4]-triazolo-[1,3,5]-triazine, was purified by column chromatography using a mixture of petroleum ether and ethyl acetate (3:1). T_dec_: 152.1 °C. FTIR (ν_max_/cm^−1^): 1602 (vs), 1481 (m), 1313 (m), 1191 (s). ^1^H NMR (400 MHz, DMSO-d_6_) δ 8.00 (dt, J = 6.6, 1.7 Hz, 6H, Ar), 7.75–7.57 (m, 9H, Ar). ^13^C NMR (400 MHz, DMSO-d_6_): *δ* 125.12, 129.00, 130.31, 131.93, 142.41 and 149.60. (Appendix A).

Synthesis of TTTL. In a three-neck flask, 0.715 g of 2-cyanophenol, 0.963 g of NH_4_Cl, 25 mL of DMF, and 1.17 g of NaN_3_ were sequentially added and heated to 130 °C for 10 h. The resulting solution was quenched in 100 mL of water, acidified to pH 1 with 6 mol/L HCl, and recrystallized in water to obtain 5-(2-hydroxyphenyl)-1H-tetrazole. In another three-neck flask, 1 g of 5-(2-hydroxyphenyl)-1H-tetrazole was dissolved in 16 mL of a 3 mol/L NaOH solution, and the resultant solution was slowly added dropwise to 0 °C with stirring. Then, 0.674 g of ice-cold acetic anhydride was added and stirred for 10 min and filtered and purified by column chromatography to obtain white solid 5-(2-acetoxyphenyl)-1H-tetrazole. Subsequently, 1.84 g of 5-(2-acetoxyphenyl)-1H-tetrazole, 4.98 g of anhydrous K_2_CO_3_, 0.55 g of cyanuric chloride, and 80 mL of methanol were sequentially added to a three-neck flask, and the mixture was heated at 90 °C for 12 h. The resulting solution was filtered, and the filtrate was collected and rotary evaporated. The resulting powder was dissolved in methanol, and the pH of the reaction system was adjusted to 8–10 by adding the NaOH solution. The mixture was then stirred at room temperature for 4 h. After the reaction was complete, the pH was adjusted to weakly acidic with diluted HCl to precipitate the white solid. The solid was filtered and purified by column chromatography to obtain tri-(2-hydroxyphenyl)-[1,2,4]-triazolo-[1,3,5]-triazine (TTTL). T_dec_: 125.8 °C. FTIR (ν_max_/cm^−1^): 3413 (s), 1614 (s), 1458 (m), 1384 (m), 1226 (m). ^1^H NMR (400 MHz, DMSO-d_6_) δ 10.31 (s, 3H, OH), 7.98 (dd, J = 7.8, 1.7 Hz, 3H, Ar), 7.40 (t, J = 1.4 Hz, 3H, Ar), 7.07 (dd, J = 8.3, 1.1 Hz, 3H, Ar), 7.00 (d, J = 1.2 Hz, 3H, Ar). ^13^C NMR (400 MHz, DMSO-d_6_): δ 113.40, 116.43, 119.25, 131.89, 133.14, 141.33, 147.11, and 157.47. (Appendix A).

Synthesis of TTTJ. The synthetic routes for TTTJ and TTTL are very similar, and as such, they are not repeated here. T_dec_: 127.5 °C. FTIR (ν_max/_cm^−1^): 3411 (s), 1612 (m), 1458 (m), 1386 (w), 1228 (w). ^1^H NMR (400 MHz, DMSO-d_6_) δ 9.94 (s, 3H, OH), 7.82 (d, J = 8.7 Hz, 3H, Ar), 7.42 (m, 3H, Ar), 7.06 (dd, J = 7.9, 1.2 Hz, 3H, Ar), 6.98 (d, J = 8.8 Hz, 3H, Ar). ^13^C NMR (400 MHz, DMSO-d_6_): δ 117.01, 120.98, 126.18, 130.14, 142.27, 149.57, 157.62, and 160.63. (Appendix A).

Synthesis of TTTD. The synthetic routes for TTTD and TTTL are very similar, and as such, they are not repeated here. T_dec_: 120.3 °C. FTIR (ν_max/_cm^−1^): 3381 (s), 1614 (vs), 1477 (s), 1396 (m), 1247 (m). ^1^H NMR (400 MHz, DMSO-d_6_) δ 10.19 (s, 3H, OH), 7.88 (d, J = 8.67 Hz, 6H, Ar), 6.96 (d, J = 8.68 Hz, 6H, Ar). ^13^C NMR (400 MHz, DMSO-d_6_): δ 111.09, 116.81, 120.15, 129.46, 133.02, and 155.79. (Appendix A).

### 3.4. Preparation of PVC/TTTs Composite Films

PVC and TTTs were dissolved in 30 mL of DMF to a total mass of 1 g (Figure 11). The mass fractions of triphenyl-[1,2,4]-triazole-[1,3,5]-triazine were 0.5 wt%, 1 wt%, and 1.5 wt%, respectively. Stir magnetons were added, and the mixture was stirred at 40 °C for 2 h. The mixture was then cast onto a smooth, clean glass plate using solvent casting before being dried in a vacuum drying oven at 80 °C for 8 h, followed by 40 °C for 4 h. The films were subsequently dipped in deionized water to remove any excess solvent and were vacuum-dried at 40 °C for 12 h.

## 4. Conclusions

In summary, we optimized the structures of four outstanding tri[1,2,4]-triazolo[1,3,5]-triazine derivatives, namely TTTB, TTTD, TTTJ, and TTTL, through theoretical calculations. Polyvinyl chloride-based UV absorbers were synthesized via cycloaddition reactions under mild conditions. These four UV absorbers were soluble in common organic solvents and could be added to PVC to resist UV radiation. To evaluate their UV absorption performance, we added these UV absorbers to the PVC, using the solvent casting method, and tested the resulting films for UV aging. The results showed that all four UV absorbers (TTTB, TTTD, TTTJ, and TTTL) exhibited an excellent UV absorption performance, effectively blocking the harmful effects of UV radiation while also having the advantages of low cost and high processability, making them suitable for industrial applications. The prepared UV absorbers were optimally applied to PVC for testing, exhibiting excellent UV absorption performance, and their use could be extended to other resin materials to reduce the harm of UV radiation to polymers. The tri-[1,2,4]-triazolo[1,3,5]-triazine heterocyclic framework can be further modified by substituents to adjust the UV absorption range and intensity, thus possessing great potential in the field of UV absorption.

## Figures and Tables

**Figure 1 ijms-24-07882-f001:**
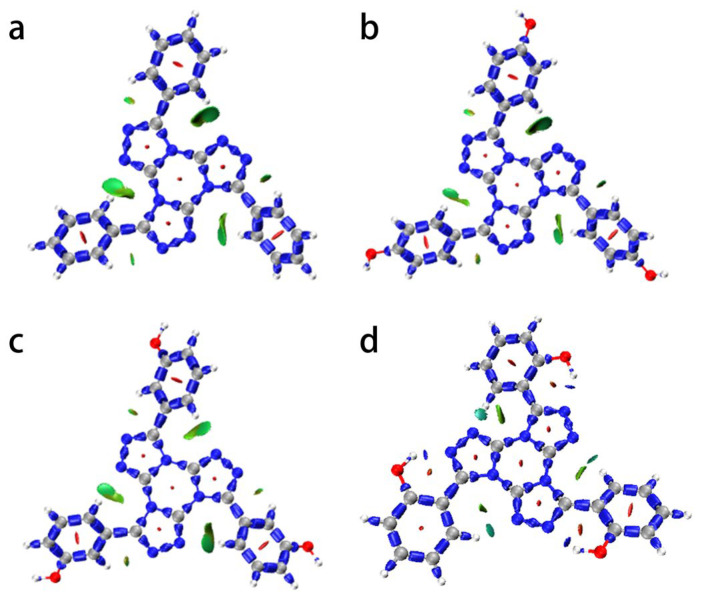
IRI contour map of TTT molecules. (**a**) TTTB; (**b**) TTTD; (**c**) TTTJ; (**d**) TTTL.

**Figure 2 ijms-24-07882-f002:**
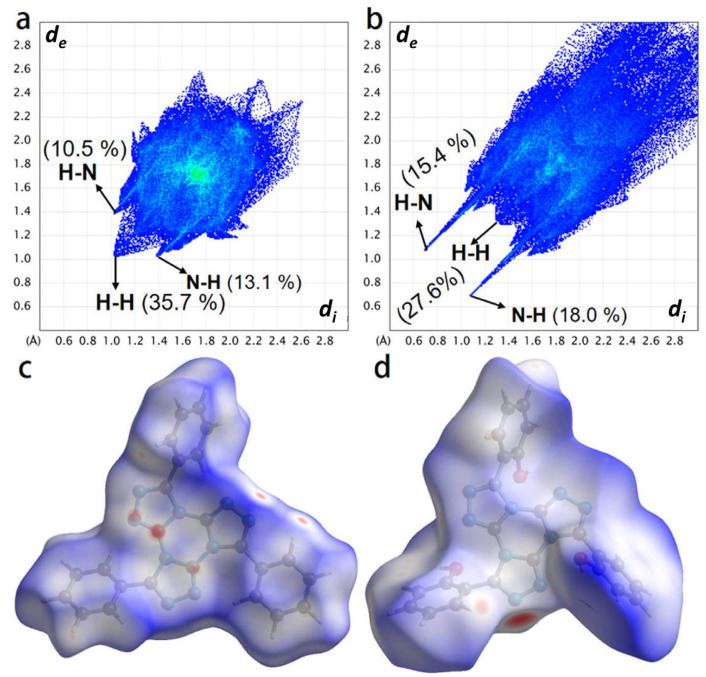
Fingerprint plots and Hirshfeld surfaces in crystal stacking for TTTB (**a**,**c**) and TTTL (**b**,**d**), respectively.

**Figure 3 ijms-24-07882-f003:**
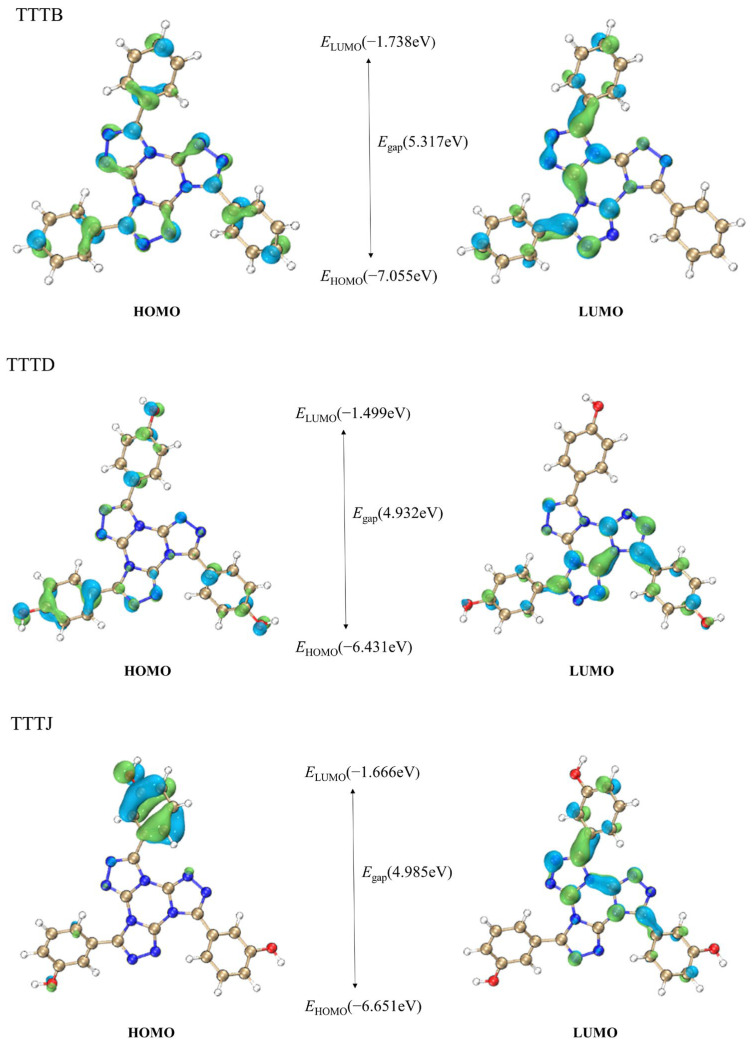
The HOMO and LUMO orbitals of compound TTTs.

**Figure 4 ijms-24-07882-f004:**
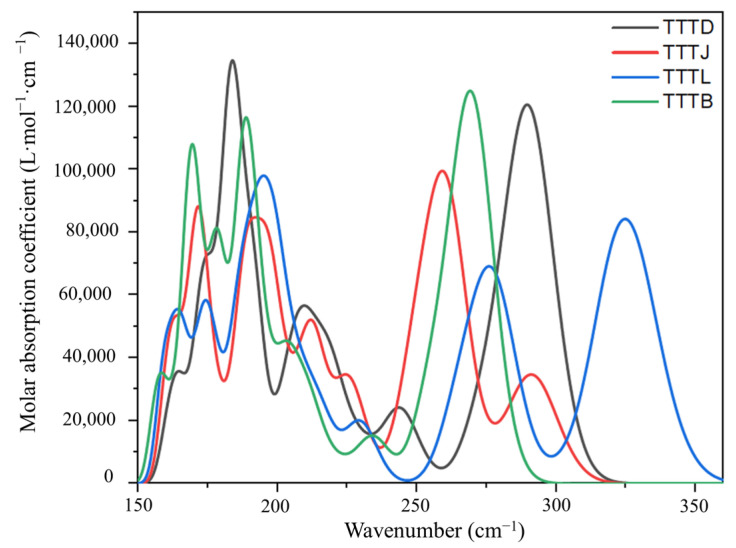
UV-vis absorption spectra of TTTB, TTTD, TTTJ, and TTTL in methanol were calculated.

**Figure 5 ijms-24-07882-f005:**
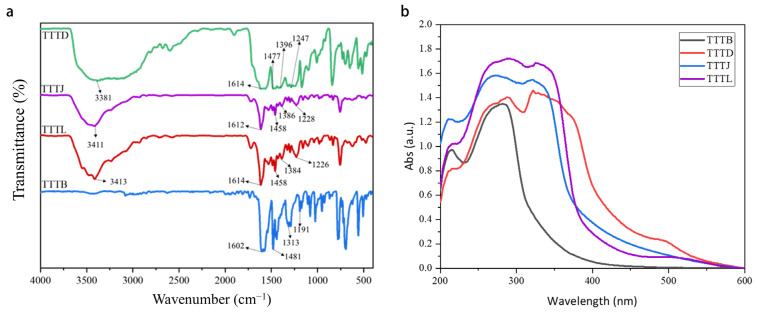
(**a**) FTIR spectrum of compounds TTTB, TTTD, TTTJ and TTTL 4000–400 cm^−1^; (**b**) Uv-vis spectra of the compounds TTTB, TTTD, TTTJ and TTTL in methanol.

**Figure 6 ijms-24-07882-f006:**
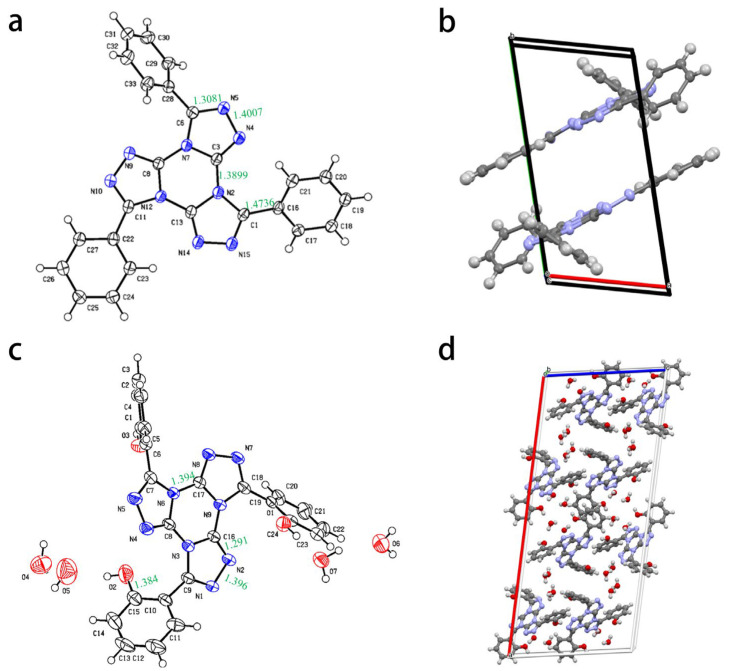
The crystals of TTTB and TTTL with their crystal packing. (**a**) The crystal structure of TTTB; (**b**) The crystal stacking structure of TTTB; (**c**) The crystal structure of TTTL; (**d**) The crystal stacking structure of TTTL.

**Figure 7 ijms-24-07882-f007:**
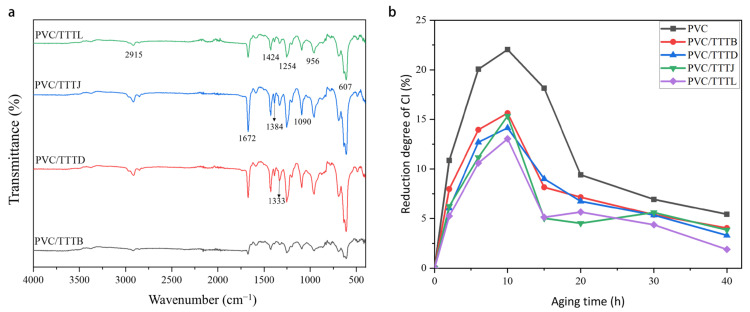
(**a**) FTIR spectra of PVC, PVC/TTTB, PVC/TTTD, PVC/TTTJ and PVC/TTTL films.; (**b**) Decreasing degree curves for the CI of PVC and PVC/TTTs films.

**Figure 8 ijms-24-07882-f008:**
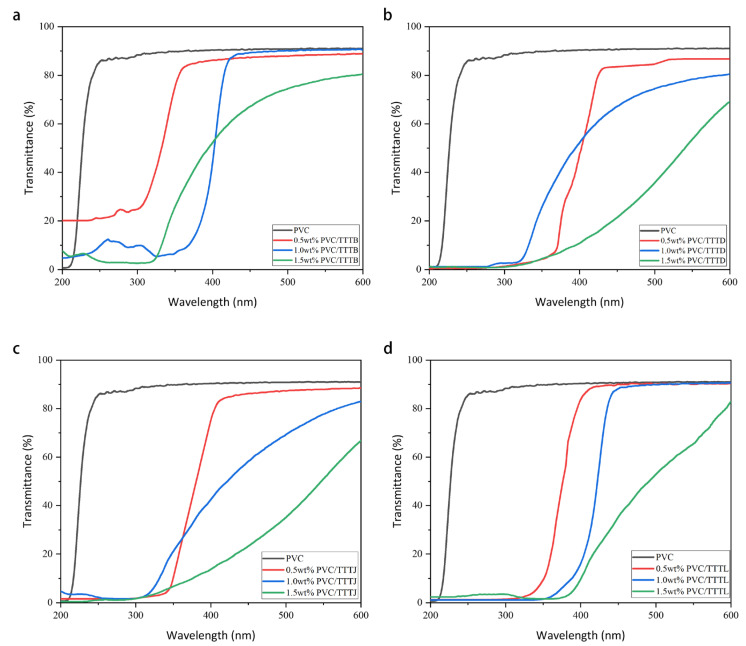
Ultraviolet-visible transmittance spectra of PVC and PVC/TTTs films with different mass fractions (**a**) TTTB; (**b**) TTTD; (**c**) TTTJ; (**d**) TTTL; (**e**) UV-0; (**f**) UV-327.

**Figure 9 ijms-24-07882-f009:**
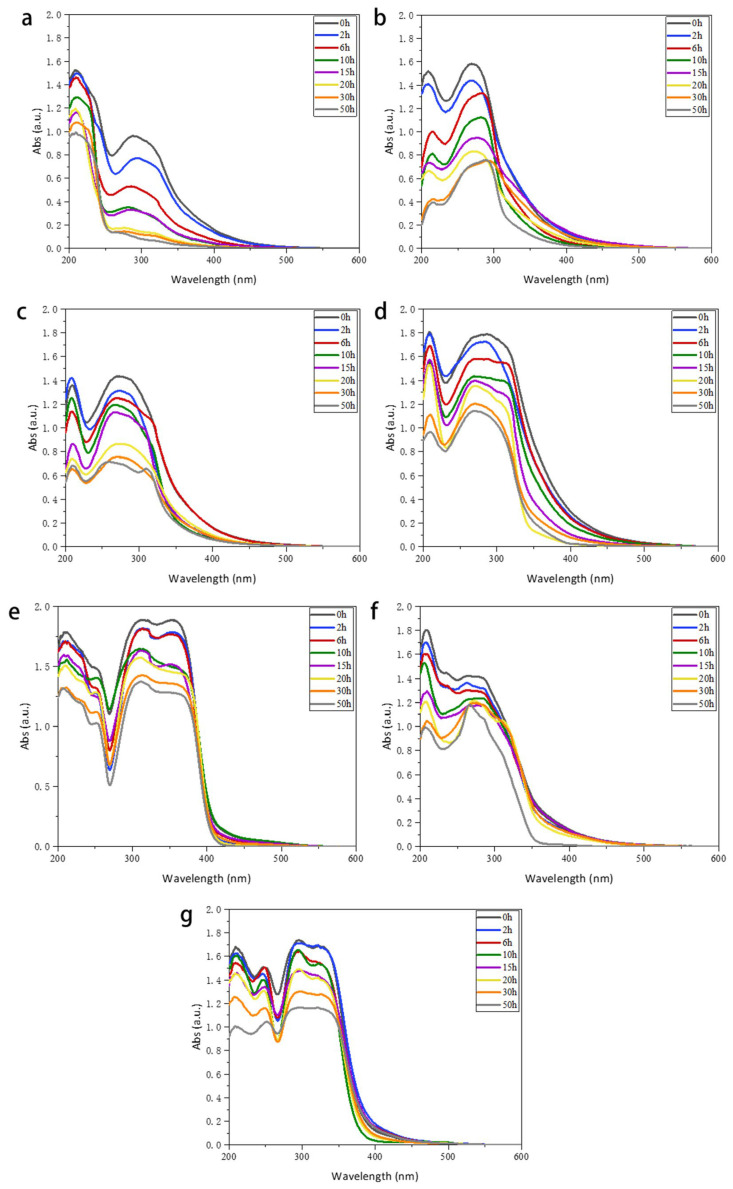
UV-vis spectra PVC films under different UV illumination times. ((**a**) PVC; (**b**) PVC/TTTB; (**c**) PVC/TTTD; (**d**) PVC/TTTJ; (**e**) PVC/TTTL; (**f**) PVC/UV-0; (**g**) PVC/UV-327).

**Figure 10 ijms-24-07882-f010:**
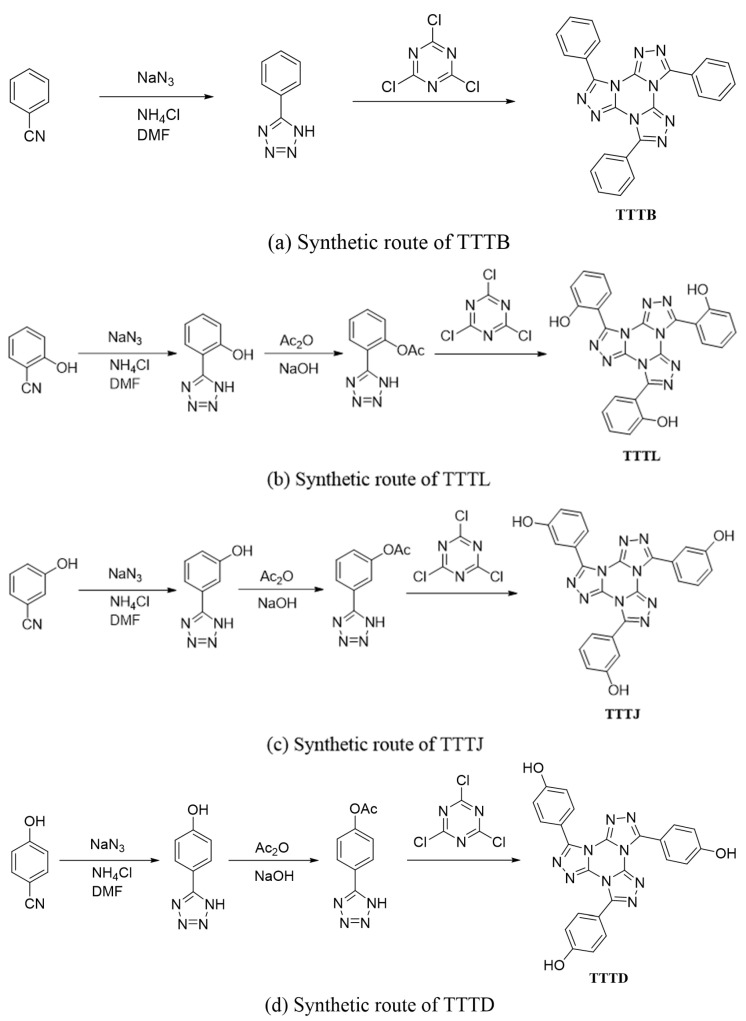
Synthetic routes of (**a**) TTTB, (**b**) TTTL, (**c**) TTTJ, (**d**) TTTD.

**Figure 11 ijms-24-07882-f011:**
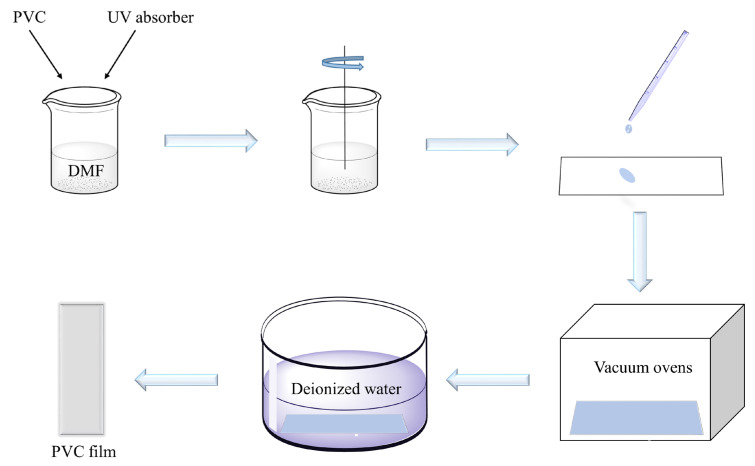
Process flowchart for preparing thin films by solvent casting method.

**Table 1 ijms-24-07882-t001:** Degradation of PVC, PVC/TTTL, PVC/TTTJ, PVC/TTTD, PVC/TTTB, PVC/UV-0 and PVC/UV-327 films at different times.

Samples	0 h	48 h	96 h	144 h
PVC	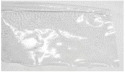	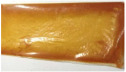	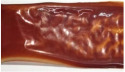	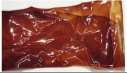
PVC/TTTL	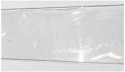	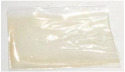	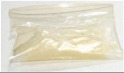	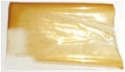
PVC/TTTJ	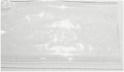	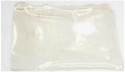	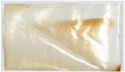	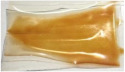
PVC/TTTD	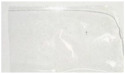	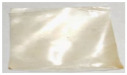	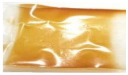	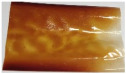
PVC/TTTB	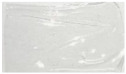	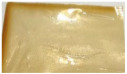	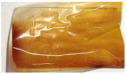	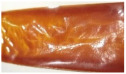
PVC/UV-0	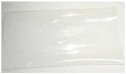	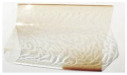	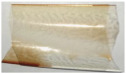	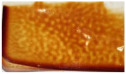
PVC/UV-327	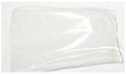	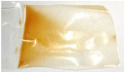	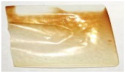	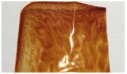

## Data Availability

Data can be found in the Appendix A.

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
