# Peer review of "The Design, Synthesis and Application of Nitrogen Heteropolycyclic Compounds with UV Resistance Properties"

_ijms, 2023, doi:10.3390/ijms24097882_

Round 1
Reviewer 1 Report
the article title The Design, Synthesis and Application of Nitrogen Heteropolycyclic Compounds with UV Resistance Properties is accepted after consideration of the following comments.
1) The rational of this study should be improved and mention the active reported compounds
2) NMR peaks should be illustrated in details .
3) discussion the chemistry of the novel compounds.
4) authors should discuss why TTTL is the most active and explain why
5) conclusion should be improved and just repeat of results
Author Response
The article title “The Design, Synthesis and Application of Nitrogen Heteropolycyclic Compounds with UV Resistance Properties” is accepted after consideration of the following comments.
- The rational of this study should be improved and mention the active reported compounds.
Response:
Thank you for pointing out the shortcomings in the manuscript. The design approach of this study is as follows: We used theoretical calculations to optimize the structure and screen out four excellent UV-absorbing tris-[1,2,4]-triazolo-[1,3,5]-triazine derivatives: TTTB, TTTD, TTTJ, and TTTL. The interaction regions indicator function, Hirshfeld surface, and 2D fingerprint analysis were used to analyze the interactions in the molecules. These four UV absorbers were successfully synthesized through a designed synthetic route, and their structures were confirmed by various testing techniques such as FTIR, 1H NMR, 13C NMR and MS. Subsequently, the characterization and aging tests of the films were carried out. They were also compared with commonly used UV absorbers in the market. We further revised and supplemented the article to improve its validity. In section 3.2.1, we mentioned the active compounds reported in 2016.
- NMR peaks should be illustrated in details.
Response:
In section 3.2.2, we have provided detailed explanations of the NMR peaks, and in the Supplementary Materials, we have identified the NMR peaks.
- Discussion the chemistry of the novel compounds.
Response:
We have included the chemical properties of the new compound, added the melting point to the preparation section, and provided a discussion on hydrogen bonding in section 3.2.5.
- Authors should discuss why TTTL is the most active and explain why.
Response:
We have supplemented explanatory information to the highlighted portions in sections 3.1 and 3.2.5.
- Conclusion should be improved and just repeat of results.
Response:
The conclusions have been revised and modified.

Reviewer 2 Report
"The Design, Synthesis and Application of Nitrogen Heteropolycyclic Compounds with UV Resistance Properties" is an original research article by Biao Yang, Xinbo Yang, Yuchuan Li and Siping Pang. The aim of the research presented in this paper, was to develop new UV resistant formulations for protection of PVC. To achieve the goal, the authors proposed a multistep synthesis of nitrogen rich compounds. They prove the structure of the end products with use of spectroscopy and XRD. They achieved high yields. The compound with the best UV shiedling properties was TTTL and the family of the compounds adds to the literature data on UV resistant formulations including organic compounds. The article is thus recommended for publication in IJMS after intriduction of minor amendments, listed below:
1. Figure 1 - the IRI abbreviation should be explained and an appropriate reference cited here. Optionally, authors could also visualise the contours of the frontier molecular orbitals.
2. Figure 2 - the appropriate citation related to Hirshfeld analysis should be placed in the caption, explaining the meaning of the d_e parameter. The software used to visualise the Hirshfeld surfaces and fingerprint plots should be cited here also.
3. Figure 3 - the part b) of the figure could be skipped, or moved to Supplementary information, as it does not bring anything valuable to understanding of the UV absorptions.
4. Figure 4 - in part a) an information is missing what kind of IR spectra were recorded: in KBr pellets or thin films ATR spectra. Moreover, the strongest absorptions should be listed in the preparation section in the format w-weak, s-strong, vs-very strong; in the part b) caption a solvent should be specified in which the spectra were recorded.
5. Figure 5 - the probability % of the thermal displacement ellipsoids should be specified in the figure caption and temperature of the measurement given in the caption. Additionally, the labels of the atoms should be given and the most important bond lengths should be listed in the figure caption. The electronic supplementary information lacks of crystallographic tables generated to describe the measurement conditions and experimental details concerning structure solution and refinement. Moreover, the numeration of compounds in the supplementary information is different to that of the main text. This should be resolved. The CIFs should be deposited in the CCDC and their numbers provided in the main text.
Author Response
“The Design, Synthesis and Application of Nitrogen Heteropolycyclic Compounds with UV Resistance Properties” is an original research article by Biao Yang, Xinbo Yang, Yuchuan Li and Siping Pang. The aim of the research presented in this paper, was to develop new UV resistant formulations for protection of PVC. To achieve the goal, the authors proposed a multistep synthesis of nitrogen rich compounds. They prove the structure of the end products with use of spectroscopy and XRD. They achieved high yields. The compound with the best uv shiedling properties was TTTL and the family of the compounds adds to the literature data on UV resistant formulations publication in IJMS after intriduction of minor amendments, listed below:
- Figure 1 - the IRI abbreviation should be explained and an appropriate reference cited here. Optionally, authors could also visualise the contours of the frontier molecular orbitals.
Response:
Thank you very much for your sincere advice. We have provided an explanation of the IRl abbreviation and cited a reference in the highlighted part of section 3.1 (line 198), and also included Figure 5 displaying the molecular frontier orbitals.
- Figure 2 - the appropriate citation related to Hirshfeld analysis should be placed in the caption, explaining the meaning of the de parameter. The software used to visualise the Hirshfeld surfaces and fingerprint plots should be cited here also.
Response:
We supplemented the relevant references, explained the meaning of the de parameter at line 216, and provided information on the software used for the Hirshfeld surface and 2D fingerprint analysis.
- Figure 3- the part b) of the figure could be skipped, or moved to Supplementary information, as it does not bring anything valuable to understanding of the UV absorptions.
Response:
We appreciate your suggestion, and we have relocated Figure 3b to the Supplementary Materials.
- Figure 4 - in part a) an information is missing what kind of IR spectra were recorded: in KBr pellets or thin films ATR spectra. Moreover, the strongest absorptions should be listed in the preparation section in the format w-weak, s-strong, vs-very strong; in the part b) caption a solvent should be specified in which the spectra were recorded.
Response:
We have provided additional information, including an explanation that Figure 4a used the potassium bromide pellet method for infrared spectroscopy analysis, supplemented the infrared data in the preparation section, and specified the solvent used for UV testing in the title.
- Figure 5- the probability % of the thermal displacement ellipsoids should be specified in the figure caption and temperature of the measurement given in the caption. Additionally, the labels of the atoms should be given and the most important bond lengths should be listed in the figure caption. The electronic supplementary information lacks of crystallographic tables generated to describe the measurement conditions and experimental details concerning structure solution and refinement. Moreover, the numeration of compounds in the supplementary information is different to that of the main text. This should be resolved. The CIFs should be deposited in the CCDC and their numbers provided in the main text. 5.
Response:
We have added information regarding the probability percentage and measurement temperature of the thermal displacement ellipsoid of the crystal at line 351, redrawn the figures with labeled atoms and important bond lengths, and provided detailed crystal information and bond lengths in Supplementary Materials S1-S4. We apologize for any confusion caused by the different compound numbering in the Supplementary Materials and the main text and have corrected the mistake. Furthermore, we have included the CCDC number of the crystal in the main text.

Reviewer 3 Report
Review of the manuscript entitled ‘The Design, Synthesis and Application of Nitrogen Heteropolycyclic Compounds with UV Resistance Properties’
Authors had UV absorber tris-[1,2,4]-triazolo-88 [1,3,5]-triazine (TTTs) to PVC and produced a film using a solvent pouring method. They submitted the PVC film to ultraviolet aging in an UV aging tester, and evaluated the film's performance.
The manuscript is interesting and deserves to be published with revisions. The introduction is good and describes the context and the goal of the study. However, the last part concerning the characterization of the thin films needs to be improved.
- The manuscript must describe how the authors have made PVC films,
- ‘Ac is the absorbance of the carbonyl vibration peak at 1715 cm-1, and Ar is the absorbance of multiple -CH2- groups at 729 cm-1.’ A sentence must be added to explain why these wavelengths have been chosen.
- The sentence: ‘Given its common use in various applications and its high susceptibility to degrada-284 tion, PVC has been selected as the primary focus of our research into developing UV-285 absorbing agents’ in page 9 must be put elsewhere at the beginning of the paragraph 2.3.
- the figure 8 and 9 are quite similar. I suggest to keep only one of these figures. The figure 9 is more explicit.
- The visual test with color is interesting must it must be coupled with a test of the rigidity of the material for example a test of the young modulus or another mechanical test.
- Materials and methods must be put at the beginning of the manuscript or in SI.
Author Response
Review of the manuscript entitled 'The Design,Synthesis and Application of Nitrogen Heteropolycyclic Compounds with uv Resistance Properties'
Authors had UV absorber tris-[1,2,4]-triazolo-[1,3,5]-triazine (TTTs) to PVC and produced a film using a solvent pouring method. They submitted the PVC film to ultraviolet aging in an UV aging tester, and evaluated the film's performance.
The manuscript is interesting and deserves to be published with revisions. The introduction is good and describes the context and the goal of the study. However, the last part concerning the characterization of the thin films needs to be improved.
- The manuscript must describe how the authors have made PVC films.
Response:
We have added a description of the process for making PVC films in section 2.4, "Preparation of PVC/TTTs composite films," and included a simplified diagram (Figure 2) to aid in understanding the steps involved in preparing the PVC films.
- – “Ac is the absorbance of the carbonyl vibration peak at 1715cm-1, and Ar is the absorbance of multiple -CH2- groups at 729cm-1.” A sentence must be added to explain why these wavelengths have been chosen.
Response:
In section 3.3.2, we have added an explanation for the selection of specific wavelengths for the UV-Vis spectroscopy measurements.
- - The sentence: 'Given its common use in various applications and its high susceptibility to degradation,PVC has been selected as the primary focus of our research into developing UV-absorbing agents' in page 9 must be put elsewhere at the beginning of the paragraph 2.3..
Response:
We have accepted your suggestion and added the information to the beginning of section 2.3.
- - the figure 8 and 9 are quite similar. I suggest to keep only one of these figures. The figure 9 is more explicit.
Response:
We have retained Figure 9 and removed Figure 8.
- The visual test with color is interesting must it must be coupled with a test of the rigidity of the material for example a test of the young modulus or another mechanical test.
Response:
Due to cracks and fractures observed in some films after the aging test, the mechanical testing results were not very satisfactory. Therefore, we provided the degradation rating of the films in Supplementary Materials S5, which can help to better understand the degree of aging of the films.
- Materials and methods must be put at the beginning of the manuscript or in SI.
Response:
We appreciate your suggestion and have revised the manuscript accordingly. The Materials and Methods section has been moved to the beginning of the manuscript for better clarity and organization.

Reviewer 4 Report
This work summarizes the Design, Synthesis, and Application of Nitrogen Heteropolycyclic Compounds with UV Resistance Properties. This work reports the chemical structure effect on the UV resistance properties. Despite the valuable results, some major issues should be addressed, and more discussion should be supplied before further consideration.
1. “The Design, Synthesis and Application of Nitrogen Heteropolycyclic Compounds with UV Resistance Properties.” Are Heteropolycyclic Compounds correct? Or Is the synthesized heterocyclic compounds?
2. In line 113, Is it intermolecular or intramolecular interaction? All figure's resolutions are too poor, and even the text is not readable.
3. Section 2.2.1 should be rephrased, and the reason should be addressed; what did the author change the condition in the recent methodology to the previous method?
4. The result and discussion of FTIR and UV-Vis are very poor. There are no explanations regarding the confirmation of organic compounds; the author addressed the data rather than explaining.
5. In Figures 4 and 6, the author should add the wavenumber to the corresponding peak. The author should add more details about the FTIR spectra of all compounds, not just the final compound (TTTB, TTTL, TTTJ, and TTTD). Further, the confirmation of all (TTTB, TTTL, TTTJ, and TTTD) compounds by NMR should be addressed briefly rather than just putting the data.
6. The FTIR explanation of the PVC films and four different PVC/TTTs films are very poor. The author should add the chemical interaction reaction of PVC/TTTs, and the FTIR explanation should be correlated with the chemical interaction of PVC/TTTs.
7. The author should add more details about all the chemicals used.
8. The author should add the representative scheme for preparing PVC/TTTs composite films.
Author Response
This work summarizes the Design, Synthesis, and Application of Nitrogen Heteropolycyclic Compounds with UV Resistance Properties. This work reports the chemical structure effect on the UV resistance properties. Despite the valuable results,some major issues should be addressed, and more discussion should be supplied before further consideration.
- “The Design,Synthesis and Application of Nitrogen Heteropolycyclic Compounds with UV Resistance Properties." Are Heteropolycyclic Compounds correct? Or Is the synthesized heterocyclic compounds?
Response:
Thank you for raising this question. After discussion and research, we have determined that the compound we prepared belongs to Heteropolycyclic Compounds, because the skeleton of the compound is a tris-[1,2,4]-triazolo-[1,3,5]-triazine fused ring system, which is a Heteropolycyclic Compound.
- In line 113, Is it intermolecular or intramolecular interaction? All figure's resolutions are too poor,and even the text is not readable.
Response:
We apologize for the mistake in line 113 and have corrected it to "intramolecular interactions". We have replaced all figures with higher resolution versions and made some language adjustments for clarity.
- Section 2.2.1 should be rephrased, and the reason should be addressed; what did the author change the condition in the recent methodology to the previous method?
Response:
We have rewritten section 2.2.1 and added information about the recent methods and their differences from the previous methods.
- The result and discussion of FTIR and UV-Vis are very poor. There are no explanations regarding the confirmation of organic compounds; the author addressed the data rather than explaining.
Response:
We have revised our discussion on the FTIR and UV-Vis results and added further explanations for the confirmation of organic compounds.
- In Figures 4 and 6, the author should add the wave number to the corresponding peak. The author should add more details about the FTIR spectra of all compounds,not just the final compound (TTTB,TTTL,TTTJ, and TTTD). Further,the confirmation of all (TTTB,TTTL,TTTJ, and TTTD) compounds by NMR should be addressed briefly rather than just putting the data.
Response:
We have added details to the FTIR spectra in Figure 2, including the wavenumbers of the peaks. Additionally, in Section 3.2.2, we have provided an explanation of the NMR data for the compound.
- The FTIR explanation of the PVC films and four different PVC/TTTs films are very poor. The author should add the chemical interaction reaction of PVC/TTTs, and the FTIR explanation should be correlated with the chemical interaction of PVC/TTTs.
Response:
We have added the wavenumbers of the peaks in the FTIR spectra of the PVC/TTTs films and provided a revised explanation. We also clarified that PVC and TTTs were only physically mixed without undergoing any chemical reactions.
- The author should add more details about all the chemicals used.
Response:
In section 2.2, we have added details about all the chemicals used in the study.
- The author should add the representative scheme for preparing PVC/TTTs composite films.
Response:
In section 2.4, "Preparation of PVC/TTTs composite films," we described how to prepare PVC films. To facilitate understanding, we have added a simplified diagram of the PVC film preparation process (Figure 2).

Round 2
Reviewer 1 Report
The article is accepted in the present form
Reviewer 4 Report
Accept